# Recreational and Nature-Based Tourism as a Cultural Ecosystem Service. Assessment and Mapping in a Rural-Urban Gradient of Central Spain

Cecilia Arnaiz-Schmitz [1,*], Cristina Herrero-Jáuregui [2] and María F. Schmitz [2]

1   Department of Civil Engineering, Transport, Territory and Urbanism, Higher Technical School of Engineering of Roads, Channels and Ports, Polytechnic University of Madrid, 28040 Madrid, Spain
2   Department of Biodiversity, Ecology and Evolution, Complutense University of Madrid, 28040 Madrid, Spain; crherrero@bio.ucm.es (C.H.-J.); ma296@ucm.es (M.F.S.)
*   Correspondence: cecilia.arnaiz@upm.es

**Abstract:** Land management focused from the social-ecological perspective of ecosystem services should consider cultural services in decision-making processes. Nature-based tourism offers a great potential for landscape conservation, local development and the well-being of human populations. However, the subjectivity of recreational ecosystem services has meant a clear impediment to assessing and mapping them. In this study, an integrated numerical spatial method is developed, which quantifies the supply and demand of recreational ecosystem services and allows mapping their spatial correspondence along a rural-urban gradient. The procedure also allows quantifying the influence of the landscape structure and the presence of protected areas on the degree of coupling between supply of recreational ecosystem services and demand for outdoor recreation and nature-based tourism and reveals that protected areas are hotspots of recreational ecosystem services. The results obtained highlight the usefulness of the methodological procedure developed as a tool for sustainable land planning and management from an integrative social-ecological approach.

**Keywords:** social-ecological systems; recreational ecosystem service hotspots; sustainable land planning and management; landscape structure; protected area effectiveness; supply-demand coupling; nature-based tourists; participatory planning procedures; local population well-being

## 1. Introduction

Cultural rural landscapes are characterised by their multi-functionality, stability and adaptation to local conditions [1]. Traditional agricultural activities have led to different types of land uses that have shaped complex landscape patterns with a high associated biocultural diversity and providing multiple ecosystem services (ES) [2]. The concept of ecosystem services links the functioning of ecosystems with social-ecological systems and human well-being [3,4]. Their supply is vulnerable to human use and therefore highly dependent on the functions of rural and urban ecosystems [5]. In this social-ecological context, the approach of rural-urban gradients is widely used to describe the relationships between land uses, urban sprawl, and ecological and socioeconomic dynamics [6–8] along with the transition from human-dominated landscapes to lightly developed rural areas [9,10]. These processes are the main driving forces of landscape structure change, impacting the supply and demand for ES. A comparison of ES supply in urban and rural regions can provide important arguments for establishing sustainable spatial planning strategies [11]. Therefore, a key challenge of land planning and management is the development of strategies based on the supply of multiple ES to control and regulate land use and urban development [12]. Likewise, the establishment of protected area networks (PAs), specially designed to preserve biodiversity and ecological flows [13], also have the potential to maintain or improve the supply of ES [14,15].

An important aspect of land management approached from the social-ecological perspective of ES is to address decision-making (the process of choosing between desirable alternatives [16]) specifically regarding what people consider valuable [17]. Exploring people's perceptions and preferences towards cultural ecosystem services (CES) can be a useful tool to identify and value those most relevant to their well-being [4,18]. This idea is proven by different authors who, through surveys, note that the values related to CES, such as recreation, cultural heritage and tourism, are among the ES considered most important by the respondents [17,19].

Cultural services are "*the non-material benefits (e.g., capabilities and experiences) that arise from human–ecosystems relationships through spiritual enrichment, cognitive development, reflection, recreation, and aesthetic experience*" [20]. Examples of CES are the appreciation of nature, landscape, culture, art and science, feelings of identity and belonging, spiritual and religious inspiration or opportunities for tourism and recreational activities, among others [21]. Nowadays, social assessment and prioritization of CES is higher than other ES due, among other reasons, to the emergence of new forms of recreation and tourism associated with aesthetic experiences and symbolic values of ecosystems [18,22–28]. Recreational and nature-based tourism (NBT), defined as "*the recreational pleasure of people derived from natural and cultural ecosystems*", not only constitutes an important type of CES, but can be a useful tool for land planning and management, nature conservation, environmental quality monitoring and an opportunity to promote local socio-economic development [23,24]. Related to this, different stakeholders have different perceptions on CES, which makes it a challenge to develop land planning strategies that incorporate all social interests. The interest in the applied use of CES and NBT makes rural areas a clear objective of sustainable land conservation and management [25,26].

Disregarding these cultural benefits in land planning, jeopardizes the effectiveness of management policies, which can be completely disconnected from the interests of society [17,29]. However, proposals to use CES in land planning and management processes face different problems and difficulties, among which the subjectivity inherent in the measurement of intangible and irreplaceable values of special importance for human well-being stands out [18,30]. Indeed, the assessment and mapping of CES is a particular challenge due to its nonmaterial value and its dependence on social constructs, among other reasons [31]. Thus, different reviews on ES quantification, modelling and mapping show that CES are the least quantified and mapped types of ES ([32,33] among others).

The general objective of this paper is to provide a numerical procedure that allows quantifying and mapping the correspondence between the supply and the demand of CES. This methodological development is intended to be useful for decision-making in the landscape planning and management from a social-ecological perspective that integrates CES. The study has focused on the potential of the landscape to supply CES related to outdoor recreation and NBT (Recreational Ecosystem Services, RES) and their demand along a rural-urban gradient in the Region of Madrid in Central Spain. For this purpose, the following specific objectives were addressed: (i) to know the landscape's potential to supply RES; (ii) to identify the demand for RES according to the perception and preferences of the visitors, collected through questionnaires; (iii) to quantify the spatial correspondence (coupling) between the supply and demand for RES along the rural-urban gradient.; (iv) to detect the relationship between the supply-demand coupling for RES, landscape structure and landscape protection; (vi) to analyse the effectiveness of PAs to provide RES.

## 2. Materials and Methods

### 2.1. Study Area

We studied a steep biophysical and socioeconomic rural-urban gradient crossing the region of Madrid (Central Spain) from Northwest to Southeast, identified in previous studies [34,35]. The study area comprises 36 municipalities covering a total of 2535 km$^2$ and its altitude varies from more than 2400 m asl in the Guadarrama mountain range, northwest of the region, to just over 400 m asl in the Jarama river valley, east of the Madrid

metropolitan area (Figure 1). This gradient is mainly characterized by pasture systems and forest-dominated landscapes in the northern uplands and agricultural systems in the southern lowlands.

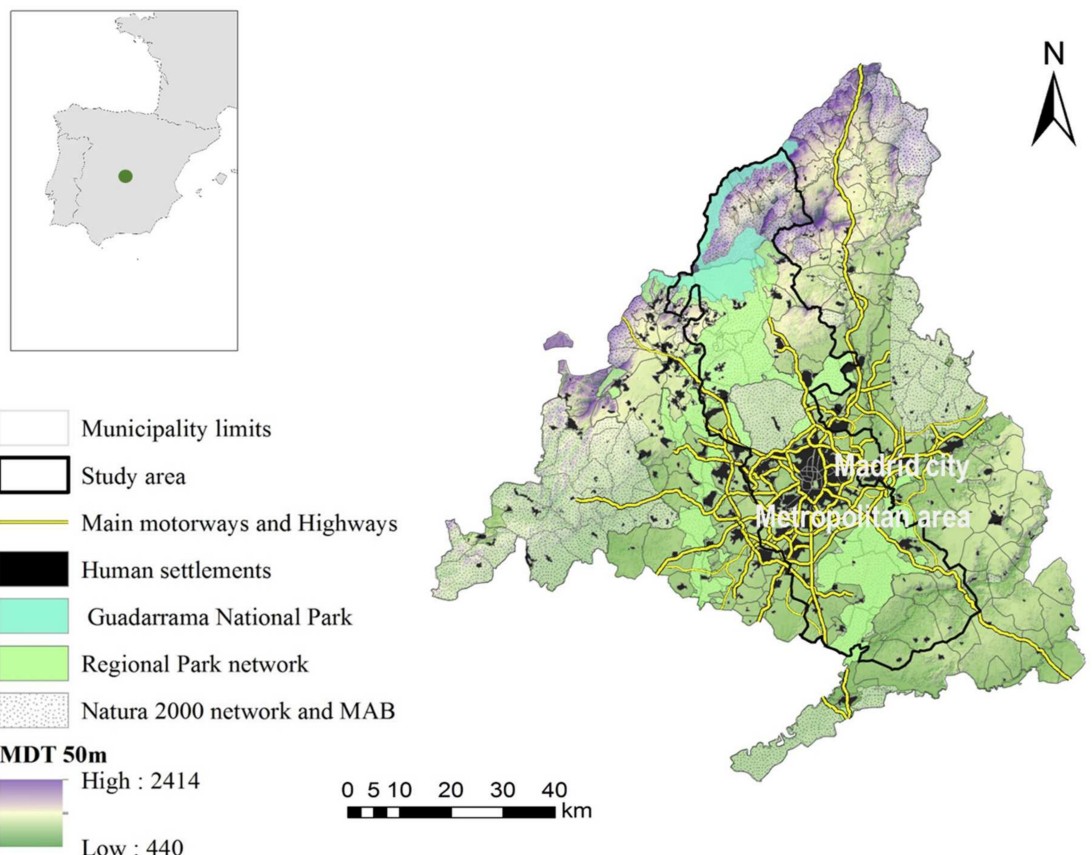

**Figure 1.** Location of the Madrid Region in Central Spain. Limits of the study area (rural-urban gradient), boundaries of the municipalities, human settlements, area occupied by the PA network, and main motorways and highways are shown. The steep altitudinal gradient of the study area was mapped with a DEM.

For several decades, the Madrid region has experienced notable socio-economic and landscape changes mainly related to intense processes of urban growth, urban sprawl [36] and rural abandonment, linked to the decrease of the population dedicated to agriculture and livestock and with the increase of employment in the secondary and tertiary sectors [34,37]. The widespread abandonment of agricultural and silvopastoral land uses and practices has favored a remarkable process of shrub encroachment and afforestation in rural areas [1,2,38]. These two main trends of landscape change, urban expansion and rurality loss, have caused an accentuated rural-urban dichotomy and significant social-ecological modifications in the study region [35]. However, in this transformed and dynamic territorial matrix, large areas with high natural and cultural values still persist, which have been recognized both regionally, nationally and internationally. Thus, a complex PA network that includes different protection categories has been established in the rural-urban gradient studied, covering one-third of the area (Figure 1): (i) two Regional Parks ("Upper Manzanares River Basin Regional Park" and "Southeast Regional Park"), a protection status recognized by the Madrid Regional Government similar to the Category V of the International Union for Conservation of Nature (IUCN), aimed at protecting and conserving both natural values and those created by the interaction with human beings through traditional management practices; (ii) a National Park ("Sierra de Guadarrama National Park"), declared of general interest by the Government of Spain due to the high value of its well preserved natural systems; (iii) a Biosphere Reserve ("Upper Basins of the Manzanares,

Lozoya and Guadarrama rivers"), an international category of protected area designated by UNESCO (Man and the Biosphere Program, MaB), which represents a balanced relationship between people and nature; (iv) six Nature 2000 sites (three Special Protection Areas, SPAs, designated under the Birds Directive of the European Union and three Sites of Community Importance, SCIs, determined by the Habitats Directive of the European Union). Both the protected landscapes of the study area and their surrounding lands have a great capacity to satisfy the recreational demand of NBT visitors [39].

### 2.2. Data Collection and Analyses

We use different types of data, collected both from public databases and available cartography and from questionnaires made to visitors to the area. As units of spatial analysis, we considered the 36 municipalities of the study area since they are the smallest governance unit in the Madrid Region and also the administrative scale of greater detail in which data are available from the socioeconomic and agricultural census (Refs. [1,2,34,35,40] among others).

The methodological approach was developed in accordance with the proposed objectives. Figure 2 summarizes the main steps followed, which are detailed below.

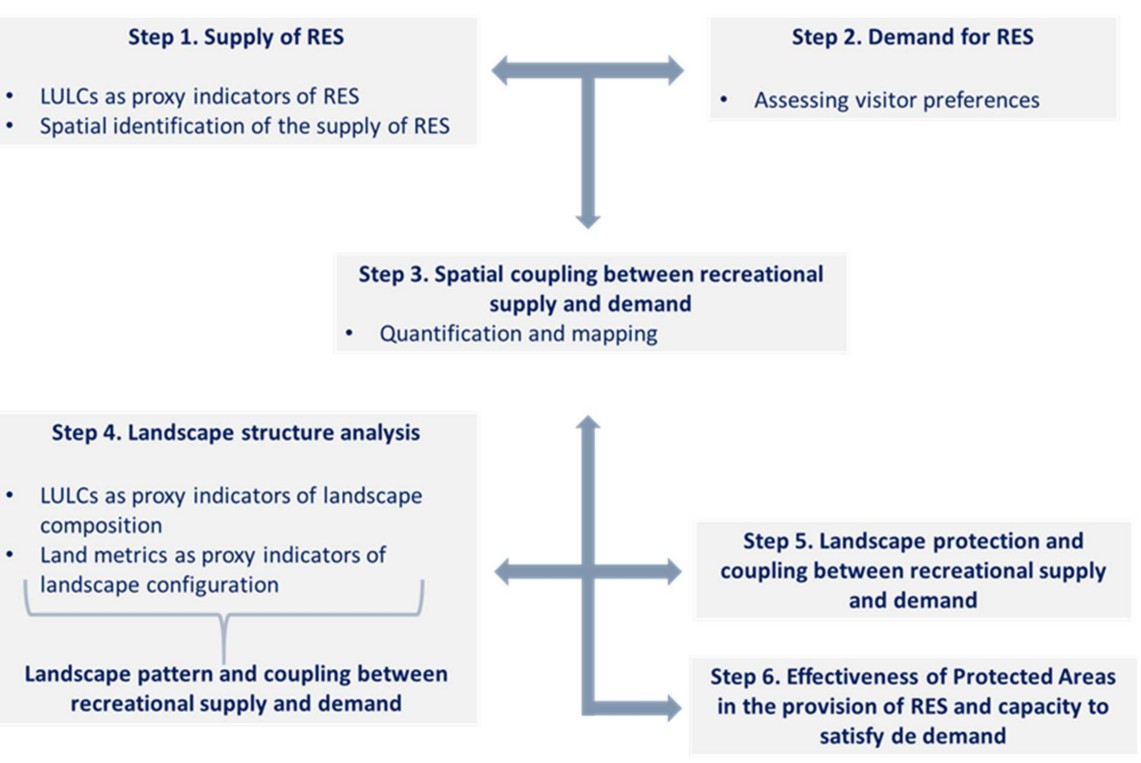

**Figure 2.** Schematic description of the steps followed in the methodological procedure.

### 2.2.1. Step 1. Identification of Nature-Based Recreational Ecosystem Services of the Landscape

The procedure followed for the assessment of the potential capacity of the land-scape to supply RES starts from the premise that the supply of ES is a function of the existing land use-land cover data (LULCs) in the territory [41]. According to several authors, we define "land cover" as the biophysical characteristics of a territory and "land use" as the utility that people obtain from it. Consequently, understanding the ways in which people use land is necessary to identify the potential supply of ES. The use of LULCs as proxies (or indirect indicators) of the ES supply is very frequent, due to the lack of empirical in-formation on ES flows [42]. Thus, proxy-based maps are more common and available that maps based on primary data [14,43]. In this work, valid and useful proxies were selected to evaluate

the capacity of the landscape to supply RES. Proxies were land uses easily recognizable by landscape observers (such as agricultural systems, grasslands, forests or shrublands, among others). These landscape features can be considered as recreational resources and potential tourism attractors [23,39] and, therefore, as ES. Examples of landscape RES are forest systems, pastures, croplands and other natural and semi-natural habitats, which provide numerous social and cultural services and represent a privileged place for outdoor recreation and leisure [44,45]. Thus, we used 16 types of LULCs for the whole rural-urban gradient, derived from the reclassification of the CORINE Land Cover data set (2012) into more significant and representative land use categories, according to the dynamics of land uses characteristic of the studied region [46] and their capacity to provide opportunities for recreational activities and NBT [21] (Table 1). In each municipality of the gradient, we calculated the spatial coverage (%) of selected LULCs, considered as recreational services. We structured this information in a data matrix in which the municipalities were row vectors that contained quantitative data about the proxies of RES (column vectors). The dimensions of the final data matrix were 36 municipalities x 16 RES proxies (Figure 3).

**Table 1.** LULCs used as proxies for recreational and nature-based tourism ecosystem services (RES) supplied by the landscape. LULCs were quantified as a percentage of occupied area in each municipality of the rural-urban gradient. These proxies were also questions in a survey on landscape preferences of visitors to the gradient studied. In this survey, visitors qualitatively indicated which elements of the landscape they found attractive.

| Recreational Service Proxies |
| --- |
| 1. **Pine forests and plantations** |
| 2. **Holm oak forests** |
| 3. **Savin juniper and Juniper forests** |
| 4. **Ash forests** |
| 5. **Lusitanian Pyrenean oak forests** |
| 6. **Mediterranean mixed forests** |
| 7. **Siliceous shrublands** |
| 8. **Kermes oak and calcicolous shrublands** |
| 9. **Grasslands** |
| 10. *Dehesa* **systems (open savannah-like woodlands used as pastures)** |
| 11. **Olive groves** |
| 12. **Rainfed agricultural land** |
| 13. **Irrigated agricultural land** |
| 14. **Crop-mosaics** |
| 15. **Riparian forests and poplar plantations** |
| 16. **Rivers, wetlands, ponds and reservoirs** |

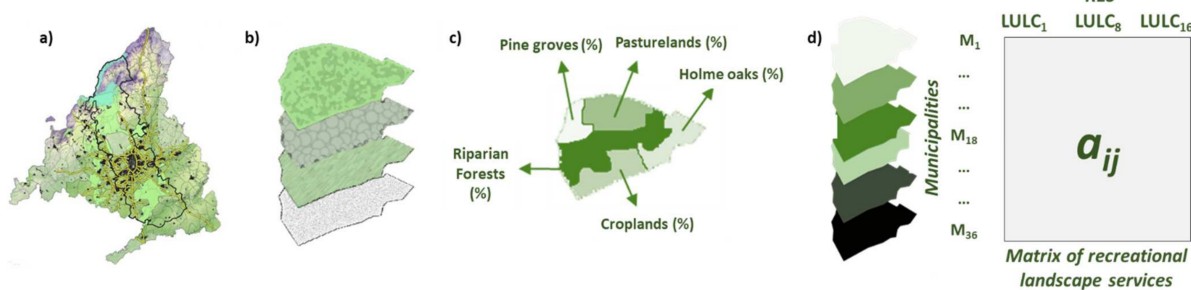

**Figure 3.** Scheme of the procedure for detecting recreational landscape services. (**a**) Spatial identification of the constituent municipalities of the rural-urban gradient studied (land units); (**b**) geospatial reclassification and selection of LULCs considered as RES proxies, at municipal level; (**c**) calculation in each municipality of the spatial cover (%) of selected LULCs; (**d**) each municipality of the rural-urban gradient is a row vector containing quantitative data (aij) of RES proxies (column data) in a matrix of recreational landscape services.



### 2.2.2. Step 2. Analysis of Outdoor Recreation Demand

LULCs, previously selected as RES proxies, were included as questions in a survey on landscape preferences of visitors to the study area. Thus, the assessment of landscape features by visitors allows to identify their RES demand [27]. The questionnaires consisted of a limited number of questions that could be easily and quickly answered and were conducted on weekends, holidays and vacation periods. The interviewees did not belong to any specific type of visitors; any visitor to the study area was considered in the survey. The respondents had to indicate which aspects of the landscape of those collected in the questionnaire (corresponding to the 16 RES selected; see Table 1) were most attractive to them, by marking the items that most closely matched their preferences. The structure and orientation of the questionnaires had been validated in previous works, with satisfactory results [22,39,47–49].

The sampling covered the heterogeneity of the landscape of the studied gradient. It was carried out along routes and at rest points in selected areas considered to be of particular interest for tourism due to their natural, recreational and historical-cultural heritage values (towns and villages, tourist information points, visitor or interpretive centres of protected areas and picnic and camping zones). To avoid possible redundancies in the responses or overrepresentations in the responses, only one person from each group of visitors was randomly chosen to fill out the questionnaire [23,39]. In the sampling, a total of 400 contacts were made, of which 367 were considered to be valid for subsequent analyses.

Quantification of demand was based on the calculation of the frequency of positive responses to the questions of the survey. Thus, we obtained a column vector containing the score of the different items of the survey, i.e., quantitative information on the demand of visitors for the RES of the landscape.

Local people were excluded from the survey because they are the "insiders", the real managers of their landscape and its tangible and intangible attributes [50]. Therefore, they were not considered in the survey so that the sample was not biased by their possible interests, problems or overvaluations [23,51,52].

### 2.2.3. Step 3. Spatial Coupling between Supply and Demand for Landscape Recreational Services. Quantification and Mapping

We used the RES supply matrix (data aij; see Figures 3d and 4a) and the demand vector to calculate the correspondence between supply and demand for recreational services (data bij) along the rural-urban gradient (Figure 4b). For this, we performed an algebraic product of matrix by vector (aij × bij). The resulting product vector informs about the degree of coupling supply-demand for RES and NBT at municipal level and its geospatial projection allowed us to obtain maps of this relationship (Figure 4c,d, respectively). The elements of this product vector (data cij) were divided into three categories (high, medium and low) by means of the natural break classification method ("Jenks optimization method" [53]), specifically designed to determine the best arrangement of values into different classes. Jenks's method is a within-groups variance minimization approach that determines class breaks to maximize homogeneity within classes and allow us to generate accurate maps, in terms of the representation of the spatial attributes of the data [54,55]. We thus obtained three types of municipalities along the rural-urban gradient, according to their greater or lesser correspondence between the supply and the demand for RES. We used ArcGis 10.6 (Esri, Redlands, CA, USA) as mapping tool.

### 2.2.4. Step 4. Analysis of the Relationship between Landscape Structure and the Supply and Demand for RES

We identified landscape structure by means of both landscape composition (using the selected LULCs, considered as valid proxies that captures the quantity of RES remaining in the landscapes; Ref. [41]) and landscape configuration (using landscape metrics, considered useful in providing an objective description of different aspects of landscape structure and patterns; Ref. [56])

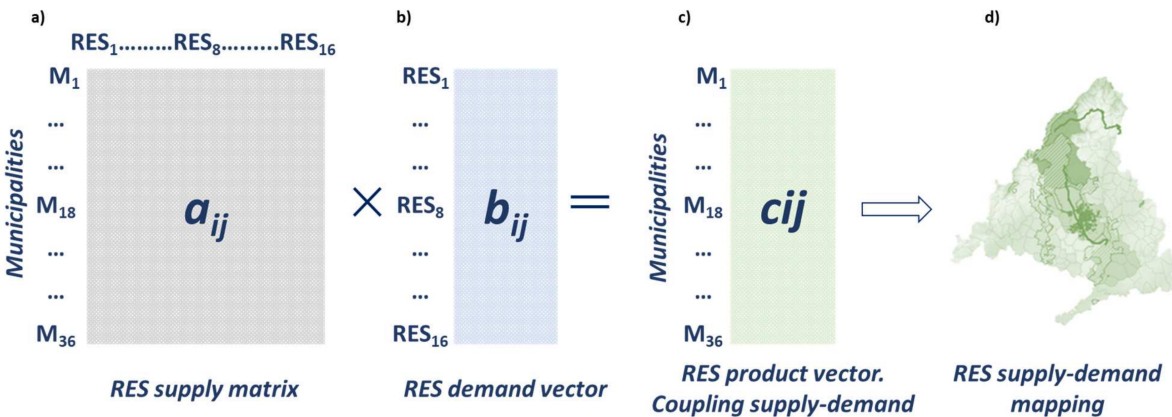

**Figure 4.** Scheme of the matrix algebra procedure followed to quantify the spatial correspondence between the supply and demand for RES. (**a**) matrix elements aij represent the supply values of RES (RES1-RES16) in each municipality (M1-M36); (**b**) vector elements bji are the demand values of RES, obtained from the calculation of the frequencies of response of the visitors to the survey questions; (**c**) output vector elements cij, obtained from a matrix-vector product, quantify the degree of coupling between the supply and demand for RES; (**d**) mapping the degree of coupling between supply and demand for RES.

Landscape metrics were selected according to criteria based on their comparability, non-redundancy, ease of interpretation, ability to describe spatial structure, and also on the satisfactory results provided by their use in previous studies [34,35,57]. The spatial metrics used were (Table 2): (1) Shannon's diversity index (SHDI, to quantify landscape diversity; it is a good indicator of landscape heterogeneity); (2) Patch richness index (PR, to measure the number of patches present); (3) Splitting index (SPLIT, to quantify landscape fragmentation [6]). Euclidean distance from the nearest neighbor (ENN, to describe the spatial isolation between patches and, consequently, the landscape connectivity). These metrics were calculated using the LULCs previously reclassified from Corine Land Cover map (2012). By means of a round moving window with a 100 m radius, we generated a raster map of each of these selected land metrics for the whole gradient studied. Subsequently, we extracted a mean value for each metric in each of the 36 municipalities included in the area. Metrics were calculated using Fragstats 4.2 [56] and were treated using ArcGis 10.6.

**Table 2.** Landscape metrics used to calculate landscape spatial patterns. The calculation procedure and a description of each metric are indicated.

| Landscape Metrics | Formula | Ranges | Description |
|---|---|---|---|
| (1) Shannon's Diversity Index | $SHDI = -\sum_{i=1}^{m}(P_i\, lnP_i)$ <br> $P_i$ = proportion of the landscape occupied by each type of patch ($i$) | $SHDI > 0$, unlimited | SHDI equals minus the sum, across all patch types, of the proportional abundance of each patch type multiplied by that proportion |
| (2) Patch richness index | $PR = m_i$ <br> $m$ = number of patch types ($i$) present in the landscape | $PR \geq 1$, unlimited | Number of different patch types present within the landscape boundary |
| (3) Splitting index | $SPLIT = \frac{A^2}{\sum_{j=1}^{n} a_{ij}^2}$ <br> $a_{ij}$ = area (m$^2$) of patch $ij$ <br> $A$ = total landscape area (m$^2$) | $1 \leq SPLIT \leq$ number cells squared in the landscape | Increases as the landscape is increasingly subdivided into smaller patches and achieves its maximum value when the landscape is maximally subdivided; that is, when every cell is a separate patch |
| (4) Euclidean nearest neighbour distance | $ENN = h_{ij}$ <br> $h_{ij}$ = distance ($m$) from patch $ij$ to nearest neighbouring patch of the same type, based on patch edge-to-edge distance computed from cell center to cell center | $ENN > 0$, unlimited | Distance to the nearest neighbouring patch of the same type, based on shortest edge-to-edge distance. It has been extensively used to quantify patch isolation |



The LULCs selected as indicators of landscape RES and the land metrics calculated were used to describe the ecologically significant characteristics of the three types of municipalities with different degrees of coupling between supply and demand, identified across the rural-urban gradient. The analysis was performed using a mean comparison test that allowed us to characterize a qualitative category (groups of municipalities) by quantitative variables (LULCs and land metrics, respectively). For this, we used a Fisher F-test (k < 2) to determine the statistical significance of the quantitative variables in the municipality types. The more the mean of a variable in a municipality type is significantly different from the mean of that variable in the whole group of municipalities, the stronger the link between the characterizing quantitative variable and the qualitative category [58]. Those groups of municipalities characterized by a high range of statistically significant variables have greater possibilities to enjoy certain land uses and recreational services.

2.2.5. Steps 5. Analysis of the Relationship between Landscape Protection and the Supply and Demand for RES

The three sets of municipalities with different coupling degree detected through the rural-urban gradient by means of the product vector elements (Figure 4), comprise inside their boundaries a dissimilar representation of the PA network established in the study area. This network is composed by several conservation categories and land-use restrictions, according to national or European legislation [35]. In these municipality types, the existence of a possible relationship between the protection of the landscape and its capacity to supply the RES demanded by the NBT visitors (degree of coupling between recreational supply and demand), was tested using Fisher's $F$ test (k < 2).

2.2.6. Step 6. Effectiveness of Protected Areas in Providing RES and Satisfying Visitors' Demand

The same test was used to detect, at landscape scale, the potential of PAs to provide a significant coupling between supply and demand for RES in the whole rural-urban gradient studied. To this end, we designed an inside/out approach, considering the PA boundaries and identifying the municipalities with a land area >25% inside the PA network and the neighboring ones, not included within its protection limits [1,2,59]. The coupling vector calculated (product vector; Figure 3c) allowed us to analyze the significant differences in the values of supply-demand correspondence inside and outside the PA network.

**3. Results**

*3.1. Landscape Potential to Supply Recreational Services*

The quantification of the selected RES proxies as a percentage of occupied land area along the rural-urban gradient, making them spatially explicit, allowed us to characterize the potential of the landscape to provide recreational services. The data matrix collecting these proxies (Figure 3d) served as the basis for the subsequent identification and characterization, through questionnaires, of the RES of greatest interest to NBT visitors, as well as to analyse and map the relationship between the recreational supply and demand of the landscape (Figure 4a).

*3.2. Visitor Preferences. Assessment of the Demand for Recreational Landscape Services*

The identification of visitor preferences was based on an analysis of the frequency of responses to a set of survey questions, from which we were able to quantify the RES of the landscape with greater weight as potential tourism attractors (Figure 4b). Positive visitor responses to each of the questions posed about the recreational services provided by the landscape often exceeded 50% (10 of the 16 questions). The most valued RES were those related to aquatic systems (riparian forests, rivers, wetlands, ponds and reservoirs), with more than 80% positive responses (Table 3).

**Table 3.** Results of the quantification of RES demand by NBT visitors. RES demand vector obtained from the calculation of the frequency of positive responses from visitors to questions in the survey about their preferences for landscape characteristics. The results are ordered from highest to lowest value of frequency of response to the survey questions.

| Survey Questions to Visitors (Indicating Their Preference for Different Aspects of the Landscape) | Demand Vector (Response Frequency, %) |
|---|---|
| Rivers, wetlands, ponds and reservoirs | 87.71 |
| Riparian forests and poplar plantations | 85.02 |
| Pine forests and plantations | 74.59 |
| Lusitanian Pyrenean oak forests | 69.38 |
| Mediterranean mixed forests | 69.05 |
| Holm oak forests | 63.19 |
| Grasslands | 60.26 |
| *Dehesas* | 58.31 |
| Siliceous shrublands | 56.03 |
| Ash forests | 50.81 |
| Kermes oak and calcicolous shrublands | 44.30 |
| Savin juniper and Juniper forests | 42.34 |
| Olive groves | 28.34 |
| Crop-mosaics | 24.43 |
| Irrigated agricultural land | 22.15 |
| Rainfed agricultural land | 14.66 |

By contrast, the least valued services were those linked to agricultural systems, which in no case reached 30% positive valuations. RES associated with pasture systems (grasslands and dehesas) and Mediterranean tree formations and shrublands occupied an intermediate position in the assessments, standing in a range of variation of positive responses between 42–75%. The vector of response frequencies thus obtained allowed us to establish the demand for each RES by visitors (Table 3).

*3.3. Measuring the Coupling between Supply and Demand for Recreational Ecosystem Services*

The spatially explicit correspondence between RES supply and demand along the rural-urban gradient was calculated by multiplying the RES supply matrix by the demand vector (Figure 4a,b). The result of this matrix operation was a vector product that allowed us to establish numerically the adjustment of the relationship between supply and demand for RES in the set of municipalities of the gradient (Figure 4c). This spatial coupling can be mapped (Figure 4d) and allows the generation of integrated RES supply-demand maps at different administrative and management scales that, in the case of this study, can vary from municipal to supra-municipal or regional scale, depending on the objectives of the land planning and management.

Figure 5 represents the spatial projection of the obtained product vector, divided into three categories of degree of coupling of RES by means of the natural break classification method [53]. The resulting types of municipalities express a clear decreasing coupling gradient, in a North-South direction, linked to the rural-urban gradient, being the municipalities located to the north of the study area those that express a greater degree of correspondence between supply and demand for recreational services. Appendix A shows the municipalities belonging to each of the three types constituted according to the degree of coupling between the supply and demand.

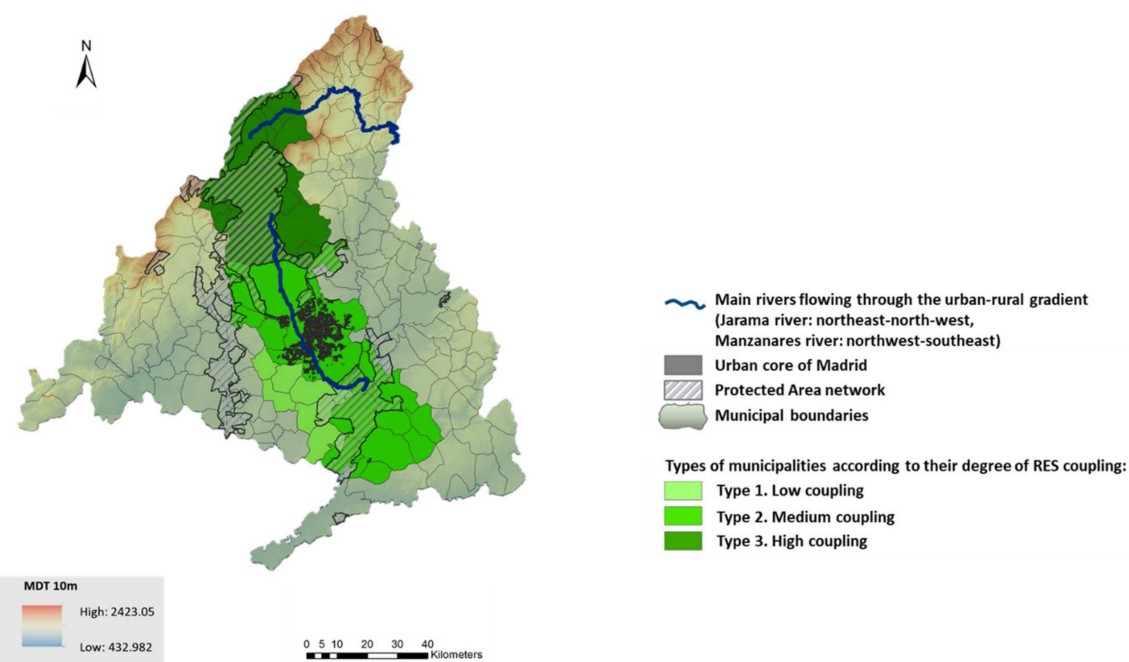

**Figure 5.** Mapping of recreational ecosystem services. Spatial projection of the types of municipalities along the rural-urban gradient studied according to their degree of coupling between the supply and demand for RES.

*3.4. Landscape Structure, Land Protection and Supply-Demand Coupling for Recreational Ecosystem Services*

We calculated the effect of the landscape structure on the coupling between recreational supply and demand considering the composition (from LULCs) and configuration (from land metrics) of the landscape of the three types of municipalities with different degrees of coupling. Fisher's *F* test highlighted a significant relationship between the variation of the landscape spatial patterns and the RES coupling along the rural-urban gradient (Table 4a).

The results obtained show that the landscape of the municipalities with a lesser degree of coupling (Type 1), located to the southwest of the study region (Figure 5), is characterized by agricultural lands, mainly rainfed crops, with a very homogeneous spatial configuration. In contrast, the group of municipalities with a higher RES coupling (Type 3), located to the north of the rural-urban gradient (Figure 5) presents significant values of richness and fragmentation of land uses and landscape heterogeneity (PRD, SPLIT and SHDI respectively; Table 4a), which results in the spatial disconnection between patches, as indicated by the Euclidean nearest neighbour distance (ENN; Table 4a). This allows to quantify the isolation of the patch (see Table 2). Statistically significant LULCs explain that Type 3 corresponds to a cultural landscape characterized by Mediterranean forest and traditional silvopastoral land uses. Type 2, with a medium degree of coupling between supply and demand for RES, is located in the centre and southeast of the study gradient (Figure 5) and presents a transitional landscape, with elements typical of both agricultural lands and Mediterranean forest systems. Landscape metrics also indicate a spatial configuration with intermediate values in relation to the patterns expressed by the types of municipalities 1 and 3. Thus, the analysis of the spatial patterns of the landscape along the rural-urban gradient has allowed us to distinguish the link between the decrease in the heterogeneity of the landscape and the degree of coupling between supply and demand for RES (i.e., the less heterogeneous the landscape, the lesser degree of coupling between the supply of RES and its demand by visitors).

**Table 4.** Characterization of the types of municipalities with different degree of coupling between recreational supply and demand, identified along the rural-urban gradient studied. Relationship between the degree of RES coupling of the municipal types and: (a) Landscape structure: (a1) LULCs, considered as proxies of RES; (a2) landscape patterns, calculated from landscape metric indices; (b) Protected area network, with different management categories and protection degree. Data are expressed as mean values per municipality types. Statistically significant values (Fisher *F*-test; *p*-value $\leq 0.05$) are indicated in bold.

| | Degree of Supply-Demand Coupling of RES | | | | | |
| | High (Landscape Type 3) | | Medium (Landscape Type 2) | | Low (Landscape Type 1) | |
| | Mean | *F*-Test | Mean | *F*-Test | Mean | *F*-Test |
|---|---|---|---|---|---|---|
| **(a) Landscape structure** | | | | | | |
| **(a1) LULCs** | | | | | | |
| *Dehesa* systems | **0.167** | 1.461 | 0.109 | −0.178 | 0.000 | −1.698 |
| Rivers, wetlands, ponds, reservoirs | **0.009** | 2.261 | 0.000 | −1.575 | 0.000 | −0.909 |
| Holm oak forests | 0.004 | −2.149 | **0.050** | 3.135 | 0.001 | −1.305 |
| Savin juniper and Juniper forests | **0.025** | 2.391 | 0.000 | −1.705 | 0.000 | −0.907 |
| Ash forests | **0.066** | 3.982 | 0.000 | −2.844 | 0.000 | −1.505 |
| Siliceous shrublands | 0.035 | −0.895 | **0.071** | 2.418 | 0.004 | −2.015 |
| Olive groves | 0.000 | −2.202 | **0.107** | 2.216 | 0.052 | −0.019 |
| Kermes oak and calcicolous shrublands | **0.136** | 2.783 | 0.049 | −1.627 | 0.024 | −1.530 |
| Grasslands | **0.194** | 4.988 | 0.005 | −1.973 | 0.000 | −3.497 |
| Pine forests and plantations | **0.225** | 3.727 | 0.050 | −2.240 | 0.006 | −1.967 |
| Riparian forests and poplar plantations | 0.023 | −2.374 | **0.026** | 1.844 | 0.002 | 0.700 |
| Lusitanian Pyrenean oak forests | **0.095** | 3.416 | 0.000 | −2.440 | 0.000 | −1.291 |
| Rainfed agricultural land | 0.000 | −4.228 | 0.212 | 0.923 | **0.515** | 4.371 |
| Irrigated agricultural land | 0.000 | −3.122 | **0.115** | 2.212 | 0.117 | 1.205 |
| Crop-mosaics | 0.000 | −2.324 | **0.082** | 2.385 | 0.038 | −0.081 |
| **(a2) Landscape metrics** | | | | | | |
| SHDI | **1.221** | 2.076 | 1.086 | −0.149 | 0.804 | −2.550 |
| PRD | **0.128** | 1.586 | 0.102 | −0.425 | 0.070 | −1.536 |
| ENN | **1.072** | 1.78 | 0.803 | −0.839 | 0.649 | −1.242 |
| SPLIT | **5.211** | 1.712 | 4.201 | −0.662 | 3.367 | −1.389 |
| **(b) Protected Area Network** | | | | | | |
| Special Protection Areas for Birds | 0.115 | −2.475 | 0.686 | 1.127 | **1.042** | 1.784 |
| Sites of Community Importance | **1.751** | 2.568 | 1.209 | −1.499 | 1.053 | −1.415 |
| Biosphere Reserve | **1.069** | 3.044 | 0.289 | −1.675 | 0.000 | −1.811 |
| Sierra de Guadarrama National Park | **0.771** | 3.807 | 0.000 | −2.719 | 0.000 | −1.439 |
| Southeast Regional Park | 0.000 | −2.809 | **0.664** | 1.941 | 0.696 | 1.148 |
| Upper Manzanares Basin Regional Park | **1.185** | 3.402 | 0.286 | −1.944 | 0.000 | −1.928 |

Likewise, the analysis of the presence of PAs in the territory corresponding to the different types of municipalities detected (number of PAs and statistical significance), showed a variation gradient of land protection in North-South direction along the study rural-urban gradient (Table 4b), linked to the decrease of the degree of coupling between landscape supply and demand for RES. Thus, the group of municipalities type 3 (maximum value of coupling) is the one that has a statistically significant greater number of PAs of different categories (Red Natura 2000, Biosphere Reserve, Regional Park and National Park). The types of municipalities 2 and 3 are characterized by comprising within their limits a Regional Park and Special Protection Areas for Birds, respectively.

In summary, through the rural-urban gradient studied we have observed the relationship between the degree of coupling of RES supply and demand with the heterogeneity of the landscape and its degree of protection. Thus, the territories with the greatest capacity for supplying the RES demanded by visitors (north of the gradient) are spatially more heterogeneous and have a greater landscape protection by means of different management strategies (Figure 6).

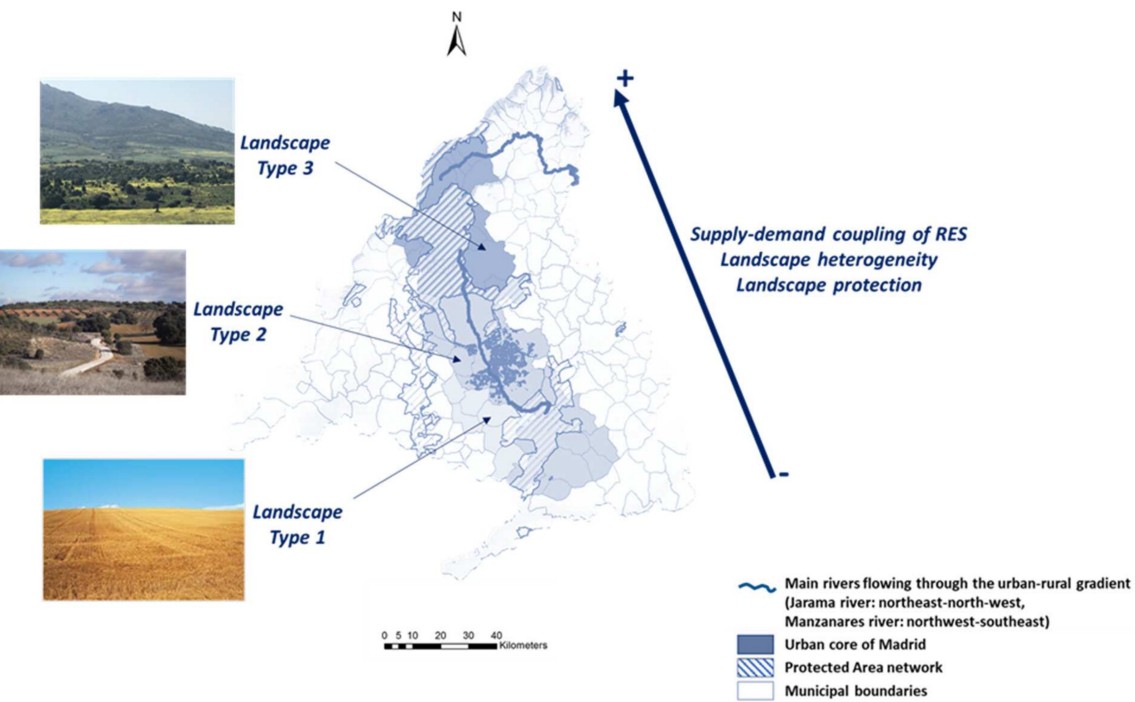

**Figure 6.** Schematic diagram summarizing the variation of the main social-ecological processes identified in the rural-urban gradient studied in the Madrid Region. The coupling between supply and demand for outdoor recreation and nature-based tourism is related to landscape heterogeneity and landscape protection.

The results of the inside/out analysis designed to test (using Fisher's *F* test) show the effectiveness of the PA network to provide a supply of RES capable of satisfying the demand of NBT visitors. The mean value of the coupling between supply and demand for recreational services inside the protected area network is statistically significant higher than outside(150.26 and 112.04, respectivelly; *p*-value $\leq 0.05$).

## 4. Discussion

Land use dynamics and management decisions are influenced by human demand for the services provided by ecosystems [60]. In this sense, changes in social demands have led to new trajectories of change in land use; such is the case of land preservation with the aim of conserving nature or providing recreational spaces [61,62]. Thus, multifunctional rural landscapes and peri-urban areas provide clear evidence of changes in land use driven by the demand for multiple ES [34]. In particular, new social demands, mainly of an increasingly urban population, have emerged for highly valued cultural services, such as outdoor recreation, NBT, protection of cultural heritage and different products that reflect the cultural identity of a region [17–19,62]. Despite this, the importance of the CES is not recognized in land use and planning schemes [63] and environmental policies do not seem to favor the participation of the population in the management of natural resources [64].

Generally, ES assessment has been performed considering the potential supply of ecosystems from a spatially explicit perspective [33]. However, few studies that assess the demand for ES (i.e., considering the amount of services required or desired by society) have modeled visitor preferences comprehensively and consistently and have produced information with spatial expression [12,65]. Furthermore, even fewer studies have analysed both supply and demand for ES from an integrated perspective [66,67]. Especially, CES are the least quantified and mapped services [68,69]. The reason for these disparities is probably related to the lack of well-established, clear and reproducible scientific methods and reference systems [60,69]. This study has among its main objectives to improve the existing methodology by making it spatially explicit.

The main contribution of the developed method is that it provides an easily replicable tool to quantify and map both the supply and demand of RES and the degree of coupling or correspondence between the set of recreational demands and the potential of the landscape to satisfy them. The method used identifies and quantifies the RES from a set of proxies based on LULCs. We selected these proxy indicators, with spatial reference, for their significant representation in the landscape and for being easily and quickly perceived by visitors to the study area [39,70]. The supply of many ES depends on their spatial context [71]. Several authors have studied the effects of the composition and configuration of the landscape on the supply of individual and multiple ES [72–75]. In this paper, we characterize the spatial configuration of the landscape using landscape metrics, which have been considered in previous analyses in the study area, as effective indicators to explore the causes and ecological meanings of landscape patterns and their consequences on ES fluxes [34,35]. The elaboration of a matrix of georeferenced data from RES has made it possible to identify and quantify the potential of the territory to provide these ecosystem services and carry out activities based on outdoor and nature tourism. The developed method facilitates the detection of possible tourist attractors on a local and regional scale.

Likewise, the demand for RES by visitors, with different perceptions and preferences, is a fundamental aspect of sustainable tourism and must be taken into account in land planning and management strategies. The survey method used was useful to determine the preferences of the visitors and their demand for RES. It is a standardized, fast, efficient and replicable technique that, considering the acceptance or rejection of landscape elements, facilitates the quantification and cartographic expression of the recreational demand [23]. The structure of the questionnaires, with a reduced number of specific questions and responses to predetermined alternatives, avoided the problems that can be caused by other approaches in which respondents must answer open-ended questions according to their own ideas. A significant number of visitors (367 people from a total of 400 contacts made) responded reliably to the survey, partly due to the structure of the questionnaire, with adequate and clear questions, easy to understand [76].

The matrix operation procedure has allowed us to quantify the correspondence or coupling between the recreational demand of visitors, based on their preferences, and the potential of the landscape to supply the required services. This spatial relationship between supply and demand for RES can be established with different degrees of adjustment [23,39]. In this work, the coupling of services has been established in a range of three categories (high, medium and low coupling), but the product vector resulting from the matrix-vector multiplication allows classifying the degree of coupling between supply and demand for RES in as many classes as observations (row vectors) compose the RES supply matrix (i.e., in this study there could be 36 coupling categories, as many as municipalities comprise the rural-urban-studied gradient). This spatial coupling can be mapped and allows the generation of maps at different administrative and management scales, depending on the objectives of land planning.

The three types of municipalities identified represent a coupling gradient of recreational supply and demand, superimposed on the rural-urban gradient studied, in the N-S direction (Figure 5). This coupling gradient is linked to the ecological characteristics of the territory and the gradual transformation of the rural landscape, with a social-ecological dynamic that generates systems in the process of rural-urban transition [34]. The characterization of these municipalities considering proxy indicators of the landscape structure and the presence of PAs within their administrative borders, has allowed to detect the gradual variation both in the spatial heterogeneity of the territory and in its level of protection (Figure 6). The landscape metrics used have proven to be valid and appropriate for landscape planning and landscape monitoring [77]. Thus, the municipalities located in the northern mountainous area of the rural-urban gradient (Sierra de Guadarrama and piedmont), mainly with silvopastoral land uses and the highest coupling values between supply and demand of RES in the study area, present a heterogeneous landscape with a high degree of protection. On the contrary, the agricultural municipalities placed at

the southern end of the rural-urban gradient have the lowest values of these landscape indicators. The identified social-ecological pattern is strongly linked to the intensity of use of the territory, whose transformation and degradation threatens the sustainability of the ES provided.

Regarding the potential of the PA network to supply RES, the inside/out approach performed shows that the PAs studied are "hotspots of outdoor recreational services", with a greater capacity to supply the services demanded by the NBT visitors than the rest of the territory. The highest coupling values between supply and demand for RES found inside the limits of the PA network support the definition of protected area by the International Union for Conservation of Nature (IUCN) as "'A clearly defined geographical space, recognized, dedicated and managed, through legal or other effective means, to achieve the long-term conservation of nature with associated ecosystem services and cultural values" [78]. In fact, PAs are considered to offer relevant opportunities to supply cultural services, such as recreation, derived mainly from their high biodiversity and degree of naturalness [79,80]. Both of them are undoubtedly important drivers of park visits.

Nevertheless, the social-ecological representativeness of PAs and the effectiveness of their management schemes are currently highly controversial. This is mainly because PA guidelines frequently inhibit some human activities, promoting land abandonment and loss of rurality, with a negative impact on the well-being of local populations, who are vulnerable to the establishment of PAs [1,2,59,81,82]. In recent decades, it is increasingly recognized that PAs must not only focus their conservation efforts on protecting wildlife and landscape naturalness, but also maintaining and supporting the livelihoods of local people by means of more effective policies addressed to protect traditional culture, because most of the PAs of the world show some degree of human use or 'culturalness' [29,83,84].

The social-ecological approach of NBT recognizes both the need to favour the quality of life of rural people and the conservation of the resources of a valuable cultural landscape, with a great potential to expand the supply of ES [28]. Recreation in protected areas is a way of engaging people with landscape conservation measures and supporting conservation goals, beyond the appreciation of biodiversity and naturalness. However, achieving a broader social commitment with conservation of PAs requires inclusive and proactive political measures that consider a wider variety of beneficiaries and the implication of different social groups through participatory planning procedures [85,86], especially local communities and their recognized traditional ecological knowledge and cultural values [29,87]. It is noteworthy that the spatial location and the boundaries of these areas do not always coincide with the perceived landscape by local people or visitors [88]. In this regard, the designed model could be applied in an inclusive context considering both the preferences of visitors to a given territory but also the opinions and attitudes of the local population towards tourism. Thus, local population must be involved in the design and management of PAs and interested in their conservation, not only for the economic benefits of NBT, but also to achieve an environmentally sustainable development that contributes to human well-being.

## 5. Conclusions

This paper provides a social-ecological framework to face outdoor recreation and NBT as a relevant representation of CES and provides an easily replicable methodological approach that favors an integrated land management, using ecological and social resources.

The method followed has allowed quantifying and mapping, in a spatially explicit way, the spatial relationships between recreational opportunities and public perceptions and preferences, determining the degree of coupling between supply and demand for recreational services. This social-ecological interaction can be expressed at multiple scales, based on different spatial reference units. In this work the municipalities of the rural-urban gradient were used as analysis units, but the methodological approach can be expressed at different administrative and supra-municipal management scales, depending on the objectives of land planning. The analysis performed has allowed to classify the rural-urban

gradient in types of social-ecological systems with different degree of coupling between supply and demand for RES and dissimilar landscape patterns as well as to detect a close relationship between degree of coupling, landscape heterogeneity and landscape protection.

The inside/out approach applied to PAs shows that they are effective hotspots of RES, with a greater capacity to supply recreational services than the rest of the territory. This proves that management guidelines of PAs and conservation-policy processes are more related to naturalness and wildlife conservation than to culturalness and the integration of local people and their cultural practices into conservation schemes. Based on these results, we emphasize the need to align nature conservation policies, tourism planning and traditional knowledge and practices, which can lead to and facilitate sustainable development.

The methodological scheme has proven to be a useful tool for strategic and sustainable land planning across rural-urban gradients and for the participatory design of PAs, generating conclusive and implementable results that are necessary in landscape management.

**Author Contributions:** Conceptualization, M.F.S. and C.A.-S.; methodology, M.F.S. and C.A.-S.; software, C.A.-S.; data curation, C.A.-S.; formal analysis, C.A.-S.; writing—original draft preparation, M.F.S. and C.A.-S.; writing—review and editing of manuscript, M.F.S., C.A.-S. and C.H.-J.; funding acquisition, M.F.S.; project administration, M.F.S. All authors have read and agreed to the published version of the manuscript.

**Funding:** This research was funded by the project LABPA-CM: Contemporary Criteria, Methods and Techniques for Landscape Knowledge and Conservation (H2019/HUM-5692), funded by the European Social Fund and the Madrid Regional Government.

**Institutional Review Board Statement:** Not applicable.

**Informed Consent Statement:** Not applicable.

**Data Availability Statement:** Not applicable.

**Acknowledgments:** The authors thank the Ministry of Science and Innovation and to the State Research Agency, the postdoctoral support provided through the Juan de la Cierva training program (FECI2019-040562-I/AEI/10.13039/50110001103).

**Conflicts of Interest:** The authors declare no conflict of interest.

## Appendix A

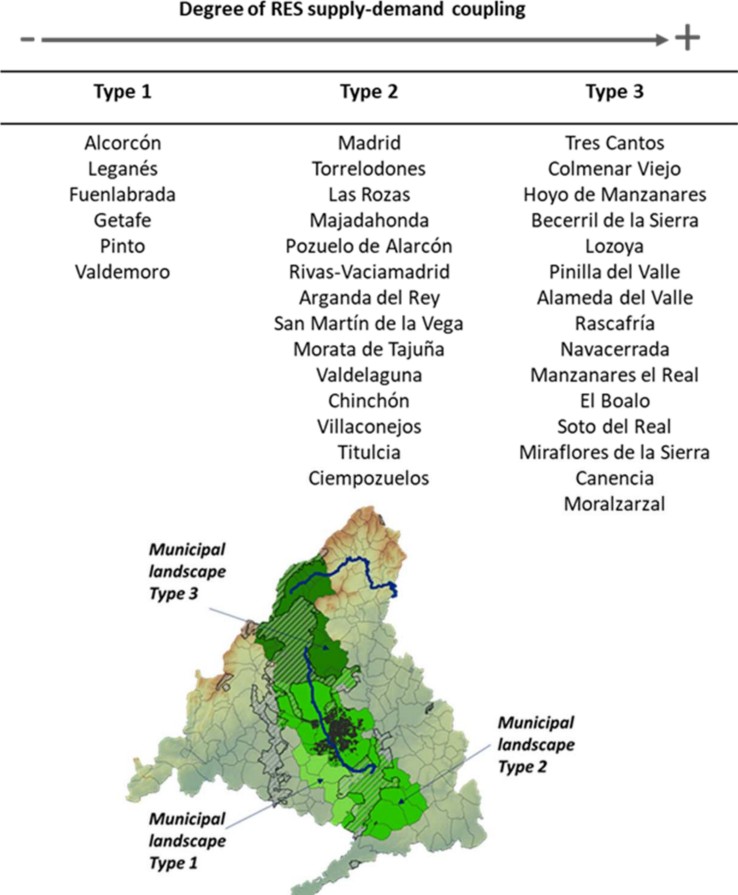

**Figure A1.** Types of municipalities according to the coupling between supply and demand of RES and their geographical location along the rural-urban gradient.

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
