# Peer review of "Recreational and Nature-Based Tourism as a Cultural Ecosystem Service. Assessment and Mapping in a Rural-Urban Gradient of Central Spain"

_land, doi:10.3390/land10040343_

Round 1

Reviewer 1 Report

In this well-written manuscript, the authors describe a novel application of a CES spatial mapping process. I see great value in their work and the technique's broader application. I have three suggestions that are more based on framing than the process, discussed below.

First – Use of urban vs rural framing

Line 31 – could you use a more specific gradient than urban to rural for ecosystems? As written suggests ES only is derived from used ecosystems.

Reading future, I think it would be worth defining what rural means in the context of this paper. So less developed, but how modified. For example, rural cropland is different than rural pasture. Both are more managed than most wild grasslands or forests.  

Second – stated vs revealed demand (and value more broadly)

Line 176 I think it would be worth noting that “demand” here is stated vs. revealed demand. This point should be made clear

Line 187 - Could the authors note how many of the 367 responses were in urban, agricultural, parks, or other spaces. Yes, the sampling occurred across the land use gradient, but would people who preferred the one type of landscape not included in the sampling points be missed?

Also, Line 56 – what do you mean by valuation and prioritization of CES is higher than other ES? Does this suggest that CES are the top value across ecosystems? I don’t think this backed up by the literature.

Line 298-300 – Perhaps resort Table 3 by percent so that it is easier to compare to the results text.  

Third – What are other tools for spatial mapping of CES. 

Could the authors compare their tool with other ways to map recreation? For example, see the link below. How is this new process additive? 

https://www.sciencedirect.com/science/article/pii/S0301479716306685?via%3Dihub

Author Response

First – Use of urban vs rural framing

Line 31 – could you use a more specific gradient than urban to rural for ecosystems? As written suggests ES only is derived from used ecosystems.

Reading future, I think it would be worth defining what rural means in the context of this paper. So less developed, but how modified. For example, rural cropland is different than rural pasture. Both are more managed than most wild grasslands or forests.

Answer: Thank you for your suggestion, we have specified at the beginning of the Introduction section the characteristics of multifunctional rural cultural landscapes and associated traditional agricultural activities and their relationship with the provision of ecosystem services and biodiversity. (lines 29-32 of the new ms)

Second – stated vs revealed demand (and value more broadly)

Line 176 I think it would be worth noting that “demand” here is stated vs. revealed demand. This point should be made clear

Answer: We have restructured the paragraph to which you refer, changing the order of the sentences, to clarify how the demand of RES has been obtained (lines 184-186 of the new ms)

Line 187 - Could the authors note how many of the 367 responses were in urban, agricultural, parks, or other spaces. Yes, the sampling occurred across the land use gradient, but would people who preferred the one type of landscape not included in the sampling points be missed?

Answer: Thanks for your comment. Certainly, the interviewed visitors were recruited along the studied gradient, covering the heterogeneity of the landscape. According to your suggestions, we have restructured the previous paragraph, noting that the survey questions referred to elements of the landscape represented by the LULC, as proxies. Therefore, all respondents answered the same questions about landscape characteristics in general and not about specific types of landscapes. This eliminates the possibility of landscapes not included in the survey (lines 184-203 of the new ms).

Also, Line 56 – what do you mean by valuation and prioritization of CES is higher than other ES? Does this suggest that CES are the top value across ecosystems? I don’t think this backed up by the literature.

Answer: Sorry, it was a mistake. The correct term is "social assessment" instead of "social valuation". We have changed it. In this sentence we highlight that the society highly values cultural services, which is widely recognized in the literature. It does not mean that the intrinsic value of these CES is greater than that of other ES. Regarding this, we have added some references in the revised manuscript. (lines 60-63 of the new ms).

Line 298-300 – Perhaps resort Table 3 by percent so that it is easier to compare to the results text.  

Answer: Thanks. We have reordered table 3 according to your suggestion (from highest to lowest percentage value)

Third – What are other tools for spatial mapping of CES. 

Could the authors compare their tool with other ways to map recreation? For example, see the link below. How is this new process additive? 

https://www.sciencedirect.com/science/article/pii/S0301479716306685?via%3Dihub

Answer: Thanks. Following your suggestion, we have specified this aspect in the revised manuscript, highlighting the scarcity of studies that have produced information with spatial expression and also that we have analysed both supply and demand for ES from an integrated perspective. Likewise, we have indicated that one of the main objectives of this work is to improve the existing methodology by making it spatially explicit. We consider that the main contribution of the developed method is that it provides an easily replicable tool to quantify and map both the supply and demand of RES and the degree of coupling or correspondence between the set of recreational demands and the potential of the landscape to satisfy them. We have clarified these points in the revised manuscript. (lines 424-437).

Reviewer 2 Report

The authors of the paper clearly identify a relevant field of research in addressing the issue of quantification and spatial representation of Recreational Ecosystem Services (RES) in the broader contexts of Nature-Based Tourism (NBT) and Cultural Ecosystem Services (CES).

They make use of a fairly common approach that entails Landscape use and Land Cover (LULC) data.

The methodology is thoroughly described, and all the passages are well identified and clear.

There are some minor concerns about some English language and style checks ( e.g., line 11 focused "on" and not "from"; line 15 "to assess" or "in Addressing"; line 19 "the supply," etc.).

One argument that the readers could raise against the foundations of the work is mainly related to the use of LULC features as proxies for RES. the issue is that LULC features by themselves are not enough to embrace all the nuances and needs in terms of features and infrastructures to provide RES aptly (e.g., point of interests, views, benches, shade, play structures, etc.). Maybe, a suggestion to approach this issue could be the insertion of a more explicatory sentence at the beginning of 2.2.1 to explain the work's perspective and its results.

Moreover, in time of COVID-19, and the following difficulties in people movements within and between countries, the decision to exclude local people from the questionnaires, even though the authors may have acquired them before the COVID crisis, should be better explained, especially in the light of the conclusions in which local communities participation is considered a staple resource to better orient management practices (in this regard, useful reference could be:

  • N. Becu, A. Neef, P. Schreinemachers, C. Sangkapitux Participatory computer simulation to support collective decision-making: potential and limits of stakeholder involvement Land Use Policy, 25 (2008), pp. 498-509
  • N. Fagerholm, N. Kayhko, F. Ndumbaro, M. Khamis Community stakeholders’ knowledge in landscape assessments −Mapping indicators for landscape services Ecol. Indic., 18 (2012), pp. 421-433
  • G. Brown, C.M. Raymond Methods for identifying land use conflict potential using participatory mapping Landscape Urban Plann., 122 (2014), pp. 196-208

).

The decision to choose only one person per group to avoid redundancy of the responses seems arbitrary since each group could be heterogeneous (adults, children, elders, differences in RES needs, etc.).

Without consideration of the local population's involvement in the management of the areas covered by the study, some of the results are unsurprising, such as the low appreciation of agricultural systems (301) or the general appreciation of more heterogeneous landscapes (377).

The work could benefit from a more extended and comprehensive definition of the designed methodology's pros and cons.

The work should be considered for publication after these minor revisions.

Author Response

  • One argument that the readers could raise against the foundations of the work is mainly related to the use of LULC features as proxies for RES. the issue is that LULC features by themselves are not enough to embrace all the nuances and needs in terms of features and infrastructures to provide RES aptly (e.g., point of interests, views, benches, shade, play structures, etc.). Maybe, a suggestion to approach this issue could be the insertion of a more explicatory sentence at the beginning of 2.2.1 to explain the work's perspective and its results.

Answer: Thank you for your comment. We have included an enlightening phrase at the beginning of the paragraph, following your suggestion. (lines 148-155)

  • Moreover, in time of COVID-19, and the following difficulties in people movements within and between countries, the decision to exclude local people from the questionnaires, even though the authors may have acquired them before the COVID crisis, should be better explained, especially in the light of the conclusions in which local communities participation is considered a staple resource to better orient management practices (in this regard, useful reference could be:
  1. Becu, A. Neef, P. Schreinemachers, C. Sangkapitux Participatory computer simulation to support collective decision-making: potential and limits of stakeholder involvement Land Use Policy, 25 (2008), pp. 498-509
  2. Fagerholm, N. Kayhko, F. Ndumbaro, M. Khamis Community stakeholders’ knowledge in landscape assessments −Mapping indicators for landscape services Ecol. Indic., 18 (2012), pp. 421-433
  3. Brown, C.M. Raymond Methods for identifying land use conflict potential using participatory mapping Landscape Urban Plann., 122 (2014), pp. 196-208).

Answer: Thanks. Indeed, the study and the surveys were carried out in a pre-Covid scenario, although their observation is very interesting and we will take it into account in future research.

We thank you for your bibliographic contributions. These have been very useful to justify the exclusion of the local population from the survey. We have added some sentences about this in section 2.2.2. (lines 208-211)

  • The decision to choose only one person per group to avoid redundancy of the responses seems arbitrary since each group could be heterogeneous (adults, children, elders, differences in RES needs, etc.)

Thank you, we understand your point of view. In reality, the objective of this premise was to avoid the over-representation of certain preferences. This design has been followed with success in previous works that now are cited in the revised manuscript. We have clarified this point in the new manuscript. (lines 199-201)

  • Without consideration of the local population's involvement in the management of the areas covered by the study, some of the results are unsurprising, such as the low appreciation of agricultural systems (301) or the general appreciation of more heterogeneous landscapes (377).

We agree. Generally, heterogeneous landscapes are preferred to agricultural ones. Beyond this interpretation, in this study landscape preferences are used as a representation of recreational demand that we have subsequently used to calculate the coupling between supply and demand for RES and its spatial projection.

  • The work could benefit from a more extended and comprehensive definition of the designed methodology's pros and cons.

Thanks. Following your suggestion, we have specified this aspect in the revised manuscript, highlighting the scarcity of studies that have produced information with spatial expression and also that we have analysed both supply and demand for ES from an integrated perspective. Likewise, we have indicated that one of the main objectives of this work is to improve the existing methodology by making it spatially explicit. We consider that the main contribution of the developed method is that it provides an easily replicable tool to quantify and map both the supply and demand of RES and the degree of coupling or correspondence between the set of recreational demands and the potential of the landscape to satisfy them. We have clarified these points in the revised manuscript. (lines 424-437)

The development method could be applied in an inclusive context considering both the preferences of visitors to a given territory but also the opinions and attitudes of the local population towards tourism. (lines 524-529)

Reviewer 3 Report

I feel that the paper is well written. But there are some points to be revised.

  1. Methodology section is too long. Could you show key points? I feel that there are too many figures in methodology section? How about unification?
  2. I feel that there are too many numbers in tables. How about changing them into graphs? And I think inside/out analysis is not necessary for your paper. Is it right? If you agree, please, delete the contents with figure 7.

Author Response

  1. Methodology section is too long. Could you show key points? I feel that there are too many figures in methodology section? How about unification?

Answer: We agree with your observation. However, we consider that unifying all the figures corresponding to the method could complicate the reading of the paper. Therefore, we have included in the manuscript Figure 2 with a schematic description of the steps followed in the methodological procedure, which allows us to synthesize and clarify the methodological development (in fact it is the last figure that we include in the paper to facilitate the understanding of the method).

2. I feel that there are too many numbers in tables. How about changing them into graphs? And I think inside/out analysis is not necessary for your paper. Is it right? If you agree, please, delete the contents with figure 7.

Answer: We are aware that Table 4 is somewhat long and contains many numbers. However, we consider that transforming this table into graphs would complicate the interpretation of the results by the reader.

The designed inside-out analysis has allowed us to know the effectiveness of the conspicuous network of protected areas existing in the study area to provide a RES offer capable of satisfying the demand of NBT visitors. The results obtained from this approach are relevant since they highlight that the PAs studied are “hotspots of outdoor recreational services”, with a greater capacity to supply the services demanded by NBT visitors than the rest of the territory (see lines 492-502). This is especially significant in an area like the one studied here, where recreation in protected areas is a way of engaging people in landscape conservation measures and supporting conservation goals. In any case, following your suggestion, we have eliminated Figure 7 and included its content in the text. (lines 406-410)